# Multi-Scale Spectral Decomposition of Massive Graphs

**Si Si**[*]
Department of Computer Science
University of Texas at Austin
ssi@cs.utexas.edu

**Donghyuk Shin**[*]
Department of Computer Science
University of Texas at Austin
dshin@cs.utexas.edu

**Inderjit S. Dhillon**
Department of Computer Science
University of Texas at Austin
inderjit@cs.utexas.edu

**Beresford N. Parlett**
Department of Mathematics
University of California, Berkeley
parlett@math.berkeley.edu

## Abstract

Computing the $k$ dominant eigenvalues and eigenvectors of massive graphs is a key operation in numerous machine learning applications; however, popular solvers suffer from slow convergence, especially when $k$ is reasonably large. In this paper, we propose and analyze a novel multi-scale spectral decomposition method (MSEIGS), which first clusters the graph into smaller clusters whose spectral decomposition can be computed efficiently and independently. We show theoretically as well as empirically that the union of all cluster's subspaces has significant overlap with the dominant subspace of the original graph, provided that the graph is clustered appropriately. Thus, eigenvectors of the clusters serve as good initializations to a block Lanczos algorithm that is used to compute spectral decomposition of the original graph. We further use hierarchical clustering to speed up the computation and adopt a fast early termination strategy to compute quality approximations. Our method outperforms widely used solvers in terms of convergence speed and approximation quality. Furthermore, our method is naturally parallelizable and exhibits significant speedups in shared-memory parallel settings. For example, on a graph with more than 82 million nodes and 3.6 billion edges, MSEIGS takes less than 3 hours on a single-core machine while Randomized SVD takes more than 6 hours, to obtain a similar approximation of the top-50 eigenvectors. Using 16 cores, we can reduce this time to less than 40 minutes.

## 1 Introduction

Spectral decomposition of large-scale graphs is one of the most informative and fundamental matrix approximations. Specifically, we are interested in the case where the top-$k$ eigenvalues and eigenvectors are needed, where $k$ is in the hundreds. This computation is needed in various machine learning applications such as semi-supervised classification, link prediction and recommender systems. The data for these applications is typically given as sparse graphs containing information about dyadic relationship between entities, e.g., friendship between pairs of users. Supporting the current big data trend, the scale of these graphs is massive and continues to grow rapidly. Moreover, they are also very sparse and often exhibit clustering structure, which should be exploited. However, popular solvers, such as subspace iteration, randomized SVD [7] and the classical Lanczos algorithm [21], are often too slow for very big graphs.

A key insight is that the graph often exhibits a clustering structure and the union of all cluster's subspaces turns out to have significant overlap with the dominant subspace of the original matrix, which

---

[*]Equal contribution to the work.

is shown both theoretically and empirically. Based on this observation, we propose a novel divide-and-conquer approach to compute the spectral decomposition of large and sparse matrices, called MSEIGS, which exploits the clustering structure of the graph and achieves faster convergence than state-of-the-art solvers. In the divide step, MSEIGS employs graph clustering to divide the graph into several clusters that are manageable in size and allow fast computation of the eigendecomposition by standard methods. Then, in the conquer step, eigenvectors of the clusters are combined to initialize the eigendecomposition of the entire matrix via block Lanczos. As shown in our analysis and experiments, MSEIGS converges faster than other methods that do not consider the clustering structure of the graph. To speedup the computation, we further divide the subproblems into smaller ones and construct a hierarchical clustering structure; our framework can then be applied recursively as the algorithm moves from lower levels to upper levels in the hierarchy tree. Moreover, our proposed algorithm is naturally parallelizable as the main steps can be carried out independently for each cluster. On the SDWeb dataset with more than 82 million nodes and 3.6 billion edges, MSEIGS takes only about 2.7 hours on a single-core machine while Matlab's `eigs` function takes about 4.2 hours and randomized SVD takes more than 6 hours. Using 16 cores, we can cut this time to less than 40 minutes showing that our algorithm obtains good speedups in shared-memory settings.

While our proposed algorithm is capable of computing highly accurate eigenpairs, it can also obtain a much faster approximate eigendecomposition with modest precision by prematurely terminating the algorithm at a certain level in the hierarchy tree. This early termination strategy is particularly useful as it is sufficient in many applications to use an approximate eigendecomposition. We apply MSEIGS and its early termination strategy to two real-world machine learning applications: label propagation for semi-supervised classification and inductive matrix completion for recommender systems. We show that both our methods are much faster than other methods while still attaining good performance. For example, to perform semi-supervised learning using label propagation on the Aloi dataset with 1,000 classes, MSEIGS takes around 800 seconds to obtain an accuracy of 60.03%; MSEIGS with early termination takes less than 200 seconds achieving an accuracy of 58.98%, which is more than 10 times faster than a conjugate gradient based semi-supervised method [10].

The rest of the paper is organized as follows. In Section 2, we review some closely related work. We present MSEIGS in Section 3 by describing the single-level case and extending it to the multi-level setting. Experimental results are shown in Section 4 followed by conclusions in Section 5.

## 2   Related Work

The spectral decomposition of large and sparse graphs is a fundamental tool that lies at the core of numerous algorithms in varied machine learning tasks. Practical examples include spectral clustering [19], link prediction in social networks [24], recommender systems with side-information [18], densest $k$-subgraph problem [20] and graph matching [22]. Most of the existing eigensolvers for sparse matrices employ the single-vector version of iterative algorithms, such as the power method and Lanczos algorithm [21]. The Lanczos algorithm iteratively constructs the basis of the Krylov subspace to obtain the eigendecomposition, which has been extensively investigated and applied in popular eigensolvers, e.g., `eigs` in Matlab (ARPACK) [14] and PROPACK [12]. However, it is well known that single-vector iterative algorithms can only compute the leading eigenvalue/eigenvector (e.g., power method) or have difficulty in computing multiplicities/clusters of eigenvalues (e.g., Lanczos). In contrast, the block version of iterative algorithms using multiple starting vectors, such as the randomized SVD [7] and block Lanczos [21], can avoid such problems and utilize efficient matrix-matrix operations (e.g., Level 3 BLAS) with better caching behavior.

While these are the most commonly used methods to compute the spectral decomposition of a sparse matrix, they do not scale well to large problems, especially when hundreds of eigenvalues/eigenvectors are needed. Furthermore, none of them consider the clustering structure of the sparse graph. One exception is the classical divide and conquer algorithm by [3], which partitions the tridiagonal eigenvalue problem into several smaller problems that are solved separately. Then it combines the solutions of these smaller problems and uses rank-one modification to solve the original problem. However, this method can only be used for tridiagonal matrices and it is unclear how to extend it to general sparse matrices.

## 3   Multi-Scale Spectral Decomposition

Suppose we are given a graph $G = (\mathcal{V}, \mathcal{E}, A)$, which consists of $|\mathcal{V}|$ vertices and $|\mathcal{E}|$ edges such that an edge between any two vertices $i$ and $j$ represents their similarity $w_{ij}$. The corresponding adjacency matrix $A$ is a $n \times n$ sparse matrix with $(i, j)$ entry equal to $w_{ij}$ if there is an edge between $i$ and $j$ and 0 otherwise. We consider the case where $G$ is an undirected graph, i.e., $A$ is symmetric. Our goal is to efficiently compute the top-$k$ eigenvalues $\lambda_1, \cdots, \lambda_k$ ($|\lambda_1| \geq \cdots \geq |\lambda_k|$) and their

corresponding eigenvectors $\mathbf{u}_1, \mathbf{u}_2, \cdots \mathbf{u}_k$ of $A$, which form the best rank-$k$ approximation of $A$. That is, $A \approx U_k \Sigma_k U_k^T$, where $\Sigma_k$ is a $k \times k$ diagonal matrix with the $k$ largest eigenvalues of $A$ and $U_k = [\mathbf{u}_1, \mathbf{u}_2, \cdots, \mathbf{u}_k]$ is an $n \times k$ orthonormal matrix. In this paper, we propose a novel multi-scale spectral decomposition method (MSEIGS), which embodies the clustering structure of $A$ to achieve faster convergence. We begin by first describing the single-level version of MSEIGS.

### 3.1 Single-level division

Our proposed multi-scale spectral decomposition algorithm, which can be used as an alternative to Matlab's `eigs` function, is based on the divide-and-conquer principle to utilize the clustering structure of the graph. It consists of two main phases: in the divide step, we divide the problem into several smaller subproblems such that each subproblem can be solved efficiently and independently; in the conquer step, we use the solutions from each subproblem as a good initialization for the original problem and achieve faster convergence compared to existing solvers which typically start from random initialization.

**Divide Step:** We first use clustering to partition the sparse matrix $A$ into $c^2$ submatrices as

$$A = D + \Delta = \begin{bmatrix} A_{11} & \cdots & A_{1c} \\ \vdots & \ddots & \vdots \\ A_{c1} & \cdots & A_{cc} \end{bmatrix}, \quad D = \begin{bmatrix} A_{11} & \cdots & 0 \\ \vdots & \ddots & \vdots \\ 0 & \cdots & A_{cc} \end{bmatrix}, \quad \Delta = \begin{bmatrix} 0 & \cdots & A_{1c} \\ \vdots & \ddots & \vdots \\ A_{c1} & \cdots & 0 \end{bmatrix}, \quad (1)$$

where each diagonal block $A_{ii}$ is a $m_i \times m_i$ matrix, $D$ is a block diagonal matrix and $\Delta$ is the matrix consisting of all off-diagonal blocks of $A$. We then compute the dominant $r$ ($r \leq k$) eigenpairs of each diagonal block $A_{ii}$ independently, such that $A_{ii} \approx U_r^{(i)} \Sigma_r^{(i)} (U_r^{(i)})^T$, where $\Sigma_r^{(i)}$ is a $r \times r$ diagonal matrix with the $r$ dominant eigenvalues of $A_{ii}$ and $U_r^{(i)} = [\mathbf{u}_1^{(i)}, \mathbf{u}_2^{(i)}, \cdots, \mathbf{u}_r^{(i)}]$ is an orthonormal matrix with the corresponding eigenvectors.

After obtaining the $r$ dominant eigenpairs of each $A_{ii}$, we can sort all $cr$ eigenvalues from the $c$ diagonal blocks and select the $k$ largest eigenvalues (in terms of magnitude) and the corresponding eigenvectors. More specifically, suppose that we select the top-$k_i$ eigenpairs of $A_{ii}$ and construct an $m_i \times k_i$ orthonormal matrix $U_{k_i}^{(i)} = [\mathbf{u}_1^{(i)}, \mathbf{u}_2^{(i)}, \cdots, \mathbf{u}_{k_i}^{(i)}]$, then we concatenate all $U_{k_i}^{(i)}$'s and form an $n \times k$ orthonormal matrix $\Omega$ as

$$\Omega = U_{k_1}^{(1)} \oplus U_{k_2}^{(2)} \oplus \cdots \oplus U_{k_c}^{(c)}, \quad (2)$$

where $\sum_i k_i = k$ and $\oplus$ denotes direct sum, which can be viewed as the sum of the subspaces spanned by $U_{k_i}^{(i)}$. Note that $\Omega$ is exactly the $k$ dominant eigenvectors of $D$. After obtaining $\Omega$, we can use it as a starting subspace for the eigendecomposition of $A$ in the conquer step. We next show that if we use graph clustering to generate the partition of $A$ in (1), then the space spanned by $\Omega$ is close to that of $U_k$, which makes the conquer step more efficient. We use principal angles [15] to measure the closeness of two subspaces. Since $\Omega$ and $U_k$ are orthonormal matrices, the $j$-th principal angle between subspaces spanned by $\Omega$ and $U_k$ is $\theta_j(\Omega, U_k) = \arccos(\sigma_j)$, where $\sigma_j$, $j = 1, 2, \cdots, k$, are the singular values of $\Omega^T U_k$ in descending order. In Theorem 3.1, we show that $\Theta(\Omega, U_k) = \text{diag}(\theta_1(\Omega, U_k), \cdots, \theta_k(\Omega, U_k))$ is related to the matrix $\Delta$.

**Theorem 3.1.** *Suppose $\lambda_1(D), \cdots, \lambda_n(D)$ (in descending order of magnitude) are the eigenvalues of $D$. Assume there is an interval $[\alpha, \beta]$ and $\eta \geq 0$ such that $\lambda_{k+1}(D), \cdots, \lambda_n(D)$ lies entirely in $[\alpha, \beta]$ and the $k$ dominant eigenvalues of $A$, $\lambda_1, \cdots, \lambda_k$, lie entirely outside of $(\alpha - \eta, \beta + \eta)$, then*

$$\| \sin(\Theta(\Omega, U_k)) \|_2 \leq \frac{\|\Delta\|_2}{\eta}, \quad \| \sin(\Theta(\Omega, U_k)) \|_F \leq \sqrt{k} \frac{\|\Delta\|_F}{\eta}.$$

The proof is given in Appendix 6.2. As we can see, $\Theta(\Omega, U_k)$ is influenced by $\Delta$, thus we need to find a partition such that $\|\Delta\|_F$ is small in order for $\| \sin(\Theta(\Omega, U_k)) \|_F$ to be small. Assuming that the graph has clustering structure, we apply graph clustering algorithms to partition $A$ to generate small $\|\Delta\|_F$. In general, the goal of graph clustering is to find clusters such that there are many edges within clusters and only a few between clusters, i.e., make $\|\Delta\|_F$ small. Various graph clustering software can be used to generate the partitions, e.g., Graclus [5], Metis [11], Nerstrand [13] and GEM [27]. Figure 1(a) shows a comparison of the cosine values of $\Theta(\Omega, U_k)$ with different $\Omega$ for the CondMat dataset, a collaboration network with 21,362 nodes and 182,628 edges. We compute $\Omega$ using random partitioning and graph clustering, where we cluster the graph into 4 clusters using Metis and more than 85% of edges appear within clusters. In Figure 1(a), more than 80% of principal angles have cosine values that are greater than 0.9 with graph clustering, whereas this ratio drops to 5% with random partitioning. This illustrates that (1) the effectiveness of graph clustering to reduce $\Theta(\Omega, U_k)$; (2) the subspace spanned by $\Omega$ from graph clustering is close to that of $U_k$.

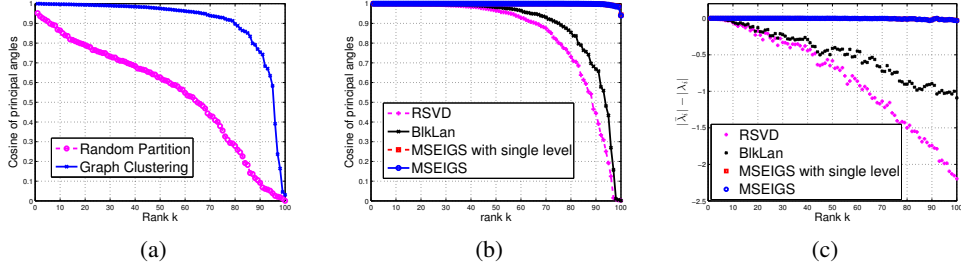

Figure 1: (a): $\cos(\Theta(\Omega, U_k))$ with graph clustering and random partition. (b) and (c): comparison of RSVD, BlkLan, MSEIGS with single level and MSEIGS on the CondMat dataset with the same number of iterations (5 steps). (b) shows $\cos(\Theta(\bar{U}_k, U_k))$, where $\bar{U}_k$ consists of the computed top-$k$ eigenvectors and (c) shows the difference between the computed eigenvalues and the exact ones.

**Conquer Step:** After obtaining $\Omega$ from the clusters (diagonal blocks) of $A$, we use $\Omega$ to initialize the spectral decomposition solver for $A$. In principle, we can use different solvers such as randomized SVD (RSVD) and block Lanczos (BlkLan). In our divide-and-conquer framework, we focus on using block Lanczos due to its superior performance as compared to RSVD. The basic idea of block Lanczos is to use an $n \times b$ initial matrix $V_0$ to construct the Krylov subspace of $A$. After $j - 1$ steps of block Lanczos, the $j$-th Krylov subspace of $A$ is given as $K_j(A, V_0) = \text{span}\{V_0, AV_0, \cdots, A^{j-1}V_0\}$. As the block Lanczos algorithm proceeds, an orthonormal basis $\hat{Q}_j$ for $K_j(A, V_0)$ is generated as well as a block tridiagonal matrix $\hat{T}_j$, which is a projection of $A$ onto $K_j(A, V_0)$. Then the Rayleigh-Ritz procedure is applied to compute the approximate eigenpairs of $A$. More details about the block Lanczos is given in Appendix 6.1. In contrast, RSVD, which is equivalent to subspace iteration with a Gaussian random matrix, constructs a basis for $A^{j-1}V_0$ and then restricts $A$ to this subspace to obtain the decomposition. As a consequence, block Lanczos can achieve better performance than RSVD with the same number of iterations.

In Figure 1(b), we compare block Lanczos with RSVD in terms of $\cos(\Theta(\bar{U}_k, U_k))$ for the CondMat dataset, where $\bar{U}_k$ consists of the approximate $k$ dominant eigenvectors. Similarly in Figure 1(c), we show that the eigenvalues computed by block Lanczos are more closer to the true eigenvalues. In other words, block Lanczos needs less iterations than RSVD to achieve similar accuracy. For the CondMat dataset, block Lanczos takes 7 iterations to achieve mean of $\cos(\Theta(\bar{U}_k, U_k))$ to be 0.99, while RSVD takes more than 10 iterations to obtain similar performance. It is worth noting that there are a number of improved versions of block Lanczos [1, 6], and we show in the experiments that our method achieves superior performance even with the simple version of block Lanczos.

The single-level version of our proposed MSEIGS algorithm is given in Algorithm 1. Some remarks on Algorithm 1 are in order: (1) $\|A_{ii}\|_F$ is likely to be different among clusters and larger clusters tend to have more influence over the spectrum of the entire matrix. Thus, we select the rank $r$ for each cluster $i$ based on the ratio $\|A_{ii}\|_F / \sum_i \|A_{ii}\|_F$; (2) We use a small number of additional eigenvectors in step 4 (similar to RSVD) to improve the effectiveness of block Lanczos; (3) It is time consuming to test convergence of the Ritz pairs in block Lanczos (steps 7, 8 of Algorithm 3 in the Appendix), thus we test convergence after running a few iterations of block Lanczos; (4) Better quality of clustering, i.e., smaller $\|\Delta\|_F$, implies higher accuracy of MSEIGS. We give performance results of MSEIGS with varying cluster quality in Appendix 6.4. From Figures 1(b) and 1(c), we can observe that the single-level MSEIGS performs much better than block Lanczos and RSVD.

We can now analyze the approximation quality of Algorithm 1 by first examining the difference between the eigenvalues computed by Algorithm 1 and the exact eigenvalues of $A$.

**Theorem 3.2.** *Let $\bar{\lambda}_1 \geq \cdots \geq \bar{\lambda}_{kq}$ be the approximate eigenvalues obtained after $q$ steps of block Lanczos in Algorithm 1. According to Kaniel-Paige Convergence Theory [23], we have*

$$\lambda_i \leq \bar{\lambda}_i \leq \lambda_i + \frac{(\lambda_1 - \lambda_i)\tan^2(\theta)}{T_{q-1}^2(\frac{1+\nu_i}{1-\nu_i})}.$$

*Using Theorem 3.1, we further have*

$$\lambda_i \leq \bar{\lambda}_i \leq \lambda_i + \frac{(\lambda_1 - \lambda_i)\|\Delta\|_2^2}{T_{q-1}^2(\frac{1+\nu_i}{1-\nu_i})(\eta^2 - \|\Delta\|_2^2)},$$

*where $T_m(x)$ is the $m$-th Chebyshev polynomial of the first kind, $\theta$ is the largest principal angle of $\Theta(\Omega, U_k)$ and $\nu_i = \frac{\lambda_i - \lambda_{k+1}}{\lambda_i - \lambda_1}$.*

Next we show the bound of Algorithm 1 in terms of rank-$k$ approximation error.

**Theorem 3.3.** *Given a $n \times n$ symmetric matrix A, suppose by Algorithm 1, we can approximate its $k$ dominant eigenpairs and form a rank-k approximation, i.e., $A \approx \bar{U}_k \bar{\Sigma}_k \bar{V}_k^T$ with $\bar{U}_k = [\bar{\boldsymbol{u}}_1, \cdots, \bar{\boldsymbol{u}}_k]$ and $\bar{\Sigma}_k = diag(\bar{\lambda}_1, \cdots, \bar{\lambda}_k)$. The approximation error can be bounded as*

$$\|A - \bar{U}_k \bar{\Sigma}_k \bar{V}_k^T\|_2 \leq 2\|A - A_k\|_2 \left(1 + \frac{\sin^2(\theta)}{1 - \sin^2(\theta)}\right)^{\frac{1}{2(q+1)}},$$

*where $q$ is the number of iterations for block Lanczos and $A_k$ is the best rank-k approximation of A. Using Theorem 3.1, we further have*

$$\|A - \bar{U}_k \bar{\Sigma}_k \bar{V}_k^T\|_2 \leq 2\|A - A_k\|_2 \left(\frac{\|\Delta\|_2^2}{\eta^2 - \|\Delta\|_2^2}\right)^{\frac{1}{2(q+1)}}.$$

The proof is given in Appendix 6.3. The above two theorems show that a good initialization is important for block Lanczos. Using Algorithm 1, we will expect a small $\|\Delta\|^2$ and $\theta$ (as shown in Figure 1(a)) because it embodies the clustering structure of $A$ and constructs a good initialization. Therefore, our algorithm can have faster convergence compared to block Lanczos with random initialization. The time complexity for Algorithm 1 is $O(|\mathcal{E}|k + nk^2)$.

---

**Algorithm 1:** MSEIGS with single level

    **Input** : $n \times n$ symmetric sparse matrix $A$, target rank $k$ and number of clusters $c$.
    **Output**: The approximate dominant $k$ eigenpairs $(\bar{\lambda}_i, \bar{\mathbf{u}}_i)$, $i = 1, \cdots, k$ of $A$.

1 Generate $c$ clusters $A_{11}, \cdots, A_{cc}$ by performing graph clustering on $A$ (e.g., Metis or Graclus).
2 Compute top-$r$ eigenpairs $(\lambda_j^{(i)}, \mathbf{u}_j^{(i)})$, $j = 1, \cdots, r$, of $A_{ii}$ using standard eigensolvers.
3 Select the top-$k$ eigenvalues and their eigenvectors from the $c$ clusters to obtain $U_{k_1}^{(1)}, \cdots, U_{k_c}^{(c)}$.
4 Form block diagonal matrix $\Omega = U_{k_1}^{(1)} \oplus \cdots \oplus U_{k_c}^{(c)}$ $(\sum_i k_i = k)$.
5 Apply block Lanczos (Algorithm 3 in Appendix 6.1) with initialization $Q_1 = \Omega$.

---

### 3.2 Multi-scale spectral decomposition

In this section, we describe our multi-scale spectral decomposition algorithm (MSEIGS). One challenge for Algorithm 1 is the trade-off in choosing the number of clusters $c$. If $c$ is large, although computing the top-$r$ eigenpairs of $A_{ii}$ can be very efficient, it is likely to increase $\|\Delta\|$, which in turn will result in slower convergence of Algorithm 1. In contrast, larger clusters will emerge when $c$ is small, increasing the time to compute the top-$r$ eigendecomposition for each $A_{ii}$. However, $\|\Delta\|$ is likely to decrease in this case, resulting in faster convergence of Algorithm 1. To address this issue, we can further partition $A_{ii}$ into $c$ smaller clusters and construct a hierarchy until each cluster is small enough to be solved efficiently. After obtaining this hierarchical clustering, we can recursively apply Algorithm 1 as it moves from lower levels to upper levels in the hierarchy tree.

By constructing a hierarchy, we can pick a small $c$ to obtain $\Omega$ with small $\Theta(\Omega, U_k)$ (we set $c = 4$ in the experiments). Our MSEIGS algorithm with multiple levels is described in Algorithm 2. Figures 1(b) and 1(c) show a comparison between MSEIGS and MSEIGS with a single level. For the single level case, we use the top-$r$ eigenpairs of the $c$ child clusters computed up to machine precision. We can see that MSEIGS performs similarly well compared to the single level case showing the effectiveness of our multi-scale approach. To build the hierarchy, we can adopt either top-down or bottom-up approaches using existing clustering algorithms. The overhead of clustering is very low, usually less than 10% of the total time. For example, MSEIGS takes 1,825 seconds, where clustering takes only 80 seconds, for the FriendsterSub dataset (in Table 1) with 10M nodes and 83M edges.

**Early Termination of MSEIGS:** Computing the exact spectral decomposition of $A$ can be quite time consuming. Furthermore, highly accurate eigenvalues/eigenvectors are not essential for many applications. Thus, we propose a fast early termination strategy (MSEIGS-Early) to approximate the eigenpairs of $A$ by terminating MSEIGS at a certain level of the hierarchy tree. Suppose that we terminate MSEIGS at the $\ell$-th level with $c_\ell$ clusters. From the top-$r$ eigenpairs of each cluster, we can select the top-$k$ eigenvalues and the corresponding eigenvectors from all $c_\ell$ clusters as an approximate eigendecomposition of $A$. As shown in Sections 4.2 and 4.3, we can significantly reduce the computation time while attaining comparable performance using the early termination strategy for two applications: label propagation and inductive matrix completion.

**Multi-core Parallelization:** An important advantage of MSEIGS is that it can be easily parallelized, which is essential for large-scale eigendecomposition. There are two main aspects of parallelism

---

**Algorithm 2:** Multi-scale spectral decomposition (MSEIGS)

---

  **Input** : $n \times n$ symmetric sparse matrix $A$, target rank $k$, the number of levels $\ell$ of the hierarchy tree and the number of clusters $c$ at each node.

  **Output**: The approximate dominant $k$ eigenpairs $(\bar{\lambda}_i, \bar{\mathbf{u}}_i)$, $i = 1, \cdots, k$ of $A$.

---

**1** Perform hierarchical clustering on $A$ (e.g., top-down or bottom-up).

**2** Compute the top-$r$ eigenpairs of each leaf node $A_{ii}^{(\ell)}$ for $i = 1, \cdots, c^\ell$, using block Lanczos.

**3** **for** $i = \ell - 1, \cdots, 1$ **do**

**4**   **for** $j = 1, \cdots, c^i$ **do**

**5**     Form block diagonal matrix $\Omega_j^{(i)}$ by (2).

**6**     Compute the eigendecomposition of $A_{jj}^{(i)}$ by Algorithm 1 with $\Omega_j^{(i)}$ as the initial block.

**7**   **end**

**8** **end**

---

in MSEIGS: (1) The eigendecomposition of clusters in the same level of the hierarchy tree can be computed independently; (2) Block Lanczos mainly involves matrix-matrix operations (Level 3 BLAS), thus efficient parallel linear algebra libraries (e.g., Intel MKL) can be used. We show in Section 4 that MSEIGS can achieve significant speedup in shared-memory multi-core settings.

## 4 Experimental Results

In this section, we empirically demonstrate the benefits of our proposed MSEIGS method. We compare MSEIGS with other popular eigensolvers including Matlab's **eigs** function (EIGS) [14], PROPACK [12], randomized SVD (RSVD) [7] and block Lanczos with random initialization (Blk-Lan) [21] on three different tasks: approximating the eigendecomposition, label propagation and inductive matrix completion. The experimental settings can be found in Appendix 6.5.

### 4.1 Approximation results

First, we show in Figure 2 the performance of MSEIGS for approximating the top-$k$ eigenvectors for different types of real-world graphs including web graphs, social networks and road networks [17, 28]. Summary of the datasets is given in Table 1, where the largest graph contains more than 3.6 billion edges. We use the average of the cosine of principal angles $\cos(\Theta(\bar{U}_k, U_k))$ as the evaluation metric, where $\bar{U}_k$ consists of the computed top-$k$ eigenvectors and $U_k$ represents the "true" top-$k$ eigenvectors computed up to machine precision using Matlab's **eigs** function. Larger values of the average $\cos(\Theta(\bar{U}_k, U_k))$ imply smaller principal angles between the subspace spanned by $U_k$ and that of $\bar{U}_k$, i.e., better approximation. As shown in Figure 2, with the same amount of time, the eigenvectors computed by MSEIGS consistently yield better principal angles than other methods.

Table 1: Datasets of increasing sizes.

| dataset | CondMat | Amazon | RoadCA | LiveJournal | FriendsterSub | SDWeb |
|---|---|---|---|---|---|---|
| # of nodes | 21,263 | 334,843 | 1,965,206 | 3,997,962 | 10.00M | 82.29M |
| # of nonzeros | 182,628 | 1,851,744 | 5,533,214 | 69,362,378 | 83.67M | 3.68B |
| rank $k$ | 100 | 100 | 200 | 500 | 100 | 50 |

Since MSEIGS divides the problem into independent subproblems, it is naturally parallelizable. In Figure 3, we compare MSEIGS with other methods under the shared-memory multi-core setting for the LiveJournal and SDWeb datasets. We vary the number of cores from 1 to 16 and show the time to compute similar approximation of the eigenpairs. As shown in Figure 3, MSEIGS achieves almost linear speedup and outperforms other methods. For example, MSEIGS is the fastest method achieving a speedup of 10 using 16 cores for the LiveJournal dataset.

### 4.2 Label propagation for semi-supervised learning and multi-label learning

One application for MSEIGS is to speed up the label propagation algorithm, which is widely used for graph-based semi-supervised learning [29] and multi-label learning [26]. The basic idea of label propagation is to propagate the known labels over an affinity graph (represented as a weighted matrix $W$) constructed using both labeled and unlabeled examples. Mathematically, at the $(t+1)$-th iteration, $F(t+1) = \alpha S F(t) + (1-\alpha)Y$, where $S$ is the normalized affinity matrix of $W$; $Y$ is the $n \times l$ initial label matrix; $F$ is the predicted label matrix; $l$ is the number of labels; $n$ is the total number of samples; $0 \leq \alpha < 1$. The optimal solution is $F^* = (1-\alpha)(I - \alpha S)^{-1}Y$. There are two standard approaches to approximate $F^*$: one is to iterate over $F(t)$ until convergence (truncated

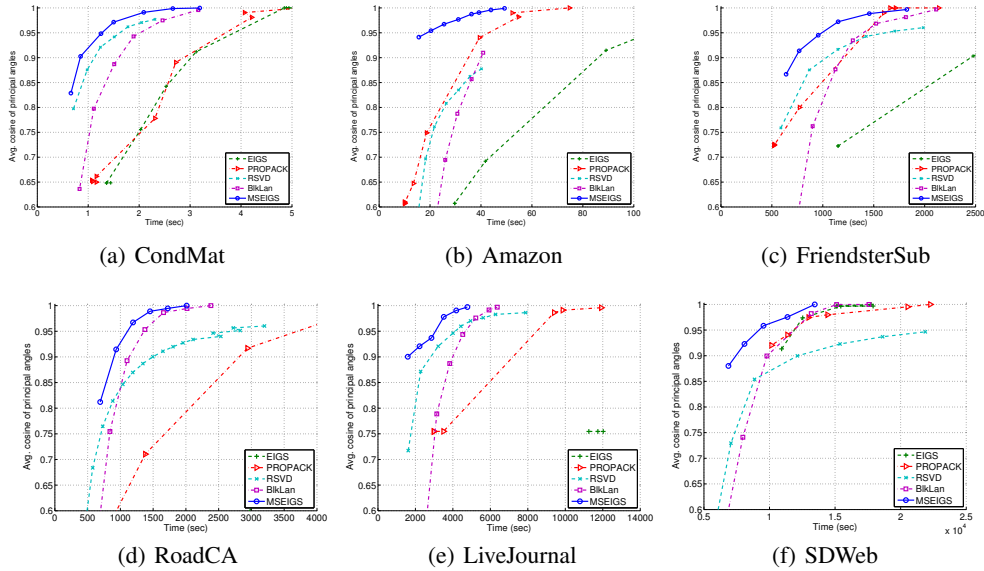

Figure 2: The $k$ dominant eigenvectors approximation results showing time vs. average cosine of principal angles. For a given time, MSEIGS consistently yields better results than other methods.

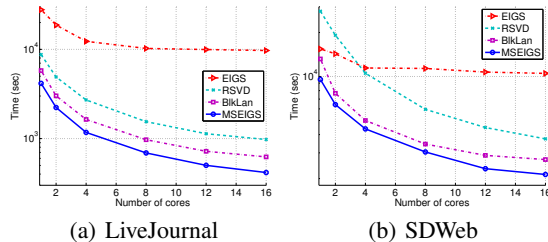

Figure 3: Shared-memory multi-core results showing number of cores vs. time to compute similar approximation. MSEIGS achieves almost linear speedup and outperforms other methods.

method); another is to solve $F^*$ as a system of linear equations by using an iterative solver like conjugate gradient (CG) [10]. However, both methods suffer from slow convergence, especially when the number of labels, i.e., columns of $Y$, grows dramatically. As an alternative, we can apply MSEIGS to generate the top-$k$ eigendecomposition of $S$ such that $S \approx \bar{U}_k \bar{\Sigma}_k \bar{U}_k^T$ and approximate $F^*$ as $F^* \approx \bar{F} = (1-\alpha)\bar{U}_k(I - \alpha\bar{\Sigma}_k)^{-1}\bar{U}_k^T Y$. Obviously, $\bar{F}$ is robust to large numbers of labels.

In Table 2, we compare MSEIGS and MSEIGS-Early with other methods for label propagation on two public datasets: Aloi and Delicious, where Delicious is a multi-label dataset containing 16,105 samples and 983 labels, and Aloi is a semi-supervised learning dataset containing 108,000 samples with 1,000 classes. More details of the datasets and parameters are given in Appendix 6.6. As we can see in Table 2, MSEIGS and MSEIGS-Early significantly outperform other methods. To achieve similar accuracy, MSEIGS takes much less time. More interestingly, MSEIGS-Early is faster than MSEIGS and almost 10 times faster than other methods with very little degradation of accuracy showing the efficiency of our early-termination strategy.

### 4.3 Inductive matrix completion for recommender systems

In the context of recommender systems, Inductive Matrix Completion (IMC) [8] is another important application where MSEIGS can be applied. IMC incorporates side-information of users and items given in the form of feature vectors for matrix factorization, which has been shown to be effective for the gene-disease association problem [18]. Given a user-item ratings matrix $R \in \mathbb{R}^{m \times n}$, where $R_{ij}$ is the known rating of item $j$ by user $i$, IMC is formulated as follows:

$$\min_{W \in \mathbb{R}^{f_c \times r}, H \in \mathbb{R}^{f_d \times r}} \sum_{(i,j) \in \Omega} (R_{ij} - \mathbf{x}_i^T W H^T \mathbf{y}_j)^2 + \frac{\lambda}{2}(\|W\|_F^2 + \|H\|_F^2),$$

where $\Omega$ is the set of observed entries; $\lambda$ is a regularization parameter; $\mathbf{x}_i \in \mathbb{R}^{f_c}$ and $\mathbf{y}_j \in \mathbb{R}^{f_d}$ are feature vectors for user $i$ and item $j$, respectively. We evaluated MSEIGS combined with IMC for recommendation tasks where a social network among users is also available. It has been shown

Table 2: Label propagation results on two real datasets including Aloi for semi-supervised classification and Delicious for multi-label learning. The graph is constructed using [16], which takes 87.9 seconds for Aloi and 16.1 seconds for Delicious. MSEIGS is about 5 times faster and MSEIGS-Early is almost 20 times faster than EIGS while achieving similar accuracy on the Aloi dataset.

| Method | Aloi ($k = 1500$) | | Delicious ($k = 1000$) | | |
|---|---|---|---|---|---|
| | time(seconds) | acc(%) | time(seconds) | top3-acc(%) | top1-acc(%) |
| Truncated | 1824.8 | 59.87 | 3385.1 | 45.12 | 48.89 |
| CG | 2921.6 | 60.01 | 1094.9 | 44.93 | 48.73 |
| EIGS | 3890.9 | 60.08 | 458.2 | 45.11 | 48.51 |
| RSVD | 964.1 | 59.62 | 359.8 | 44.11 | 46.91 |
| BlkLan | 1272.2 | 59.96 | 395.6 | 43.52 | 45.53 |
| MSEIGS | 767.1 | 60.03 | 235.6 | 44.84 | 49.23 |
| MSEIGS-Early | 176.2 | 58.98 | 61.36 | 44.71 | 48.22 |

that exploiting these social networks improves the quality of recommendations [9, 25]. One way to obtain useful and robust features from the social network is to consider the $k$ principal components, i.e., top-$k$ eigenvectors, of the corresponding adjacency matrix $A$. We compare the recommendation performance of IMC using eigenvectors computed by MSEIGS, MSEIGS-Early and EIGS. We also report results for two baseline methods: standard matrix completion (MC) without user/item features and Katz[1] on the combined network $C = [A\ R; R^T\ 0]$ as in [25].

We evaluated the recommendation performance on three publicly available datasets shown in Table 6 (see Appendix 6.7 for more details). The Flixster dataset [9] contains user-movie ratings information and the other two datasets [28] are for the user-affiliation recommendation task. We report recall-at-$N$ with $N = 20$ averaged over 5-fold cross-validation, which is a widely used evaluation metric for top-$N$ recommendation tasks [2]. In Table 3, we can see that IMC outperforms the two baseline methods: Katz and MC. For IMC, both MSEIGS and MSEIGS-Early achieve comparable results compared to other methods, but require much less time to compute the top-$k$ eigenvectors (i.e., user latent features). For the LiveJournal dataset, MSEIGS-Early is almost 8 times faster than EIGS while attaining similar performance as shown in Table 3.

Table 3: Recall-at-20 (RCL@20) and top-$k$ eigendecomposition time (eig-time, in seconds) results on three real-world datasets: Flixster, Amazon and LiveJournal. MSEIGS and MSEIGS-Early require much less time to compute the top-$k$ eigenvectors (latent features) for IMC while achieving similar performance compared to other methods. Note that Katz and MC do not use eigenvectors.

| Method | Flixster ($k = 100$) | | Amazon ($k = 500$) | | LiveJournal ($k = 500$) | |
|---|---|---|---|---|---|---|
| | eig-time | RCL@20 | eig-time | RCL@20 | eig-time | RCL@20 |
| Katz | - | 0.1119 | - | 0.3224 | - | 0.2838 |
| MC | - | 0.0820 | - | 0.4497 | - | 0.2699 |
| EIGS | 120.51 | 0.1472 | 871.30 | 0.4999 | 12099.57 | 0.4259 |
| RSVD | 85.31 | 0.1491 | 369.82 | 0.4875 | 7617.98 | 0.4294 |
| BlkLan | 104.95 | 0.1465 | 882.58 | 0.4687 | 5099.79 | 0.4248 |
| MSEIGS | 36.27 | 0.1489 | 264.47 | 0.4911 | 2863.55 | 0.4253 |
| MSEIGS-Early | 21.88 | 0.1481 | 179.04 | 0.4644 | 1545.52 | 0.4246 |

## 5 Conclusions

In this paper, we proposed a novel divide-and-conquer based framework, multi-scale spectral decomposition (MSEIGS), for approximating the top-$k$ eigendecomposition of large-scale graphs. Our method exploits the clustering structure of the graph and converges faster than state-of-the-art methods. Moreover, our method can be easily parallelized, which makes it suitable for massive graphs. Empirically, MSEIGS consistently outperforms other popular eigensolvers in terms of convergence speed and approximation quality on real-world graphs with up to billions of edges. We also show that MSEIGS is highly effective for two important applications: label propagation and inductive matrix completion. Dealing with graphs that cannot fit into memory is one of our future research directions. We believe that MSEIGS can also be efficient in streaming and distributed settings with careful implementation.

### Acknowledgments

This research was supported by NSF grant CCF-1117055 and NSF grant CCF-1320746.

## Footnotes

[1]The Katz measure is defined as $\sum_{i=1}^{t} \beta^t C^t$. We set $\beta = 0.01$ and $t = 10$.

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
