[Supplementary Material]

# 6 Appendix

## 6.1 Block Lanczos algorithm

The block Lanczos algorithm is a widely used eigendecomposition method for computing the dominant $k$ eigenvalues and eigenvectors of a symmetric matrix $A$ [21]. The basic idea of block Lanczos is to use an $n \times b$ initial matrix $V_0$ to construct the Krylov subspace of $A$. After $j - 1$ steps, the algorithm generates an orthonormal basis $\hat{Q}_j = [Q_1, Q_2, \cdots, Q_j]$ of the $j$-th Krylov subspace of $A$ as $K_j(A, V_0) = \text{span}\{V_0, AV_0, \cdots, A^{j-1}V_0\}$, which satisfies the three-term recurrence $Q_{j+1}B_j = AQ_j - Q_jA_j - Q_{j-1}B_{j-1}^T$. Simultaneously with the iteration, a sequence of block tridiagonal matrices $\hat{T}_j$ is obtained, each of which is an orthonormal projection of $A$ onto $K_j(A, V_0)$:

$$\hat{T}_j = \hat{Q}_j^T A \hat{Q}_j = \begin{bmatrix} A_1 & B_1^T & \cdots & 0 \\ B_1 & A_2 & \ddots & \vdots \\ \vdots & \ddots & \ddots & B_j^T \\ 0 & \cdots & B_j & A_{j+1} \end{bmatrix}.$$

Then, the governing equation for block Lanczos can be written as

$$A\hat{Q}_j - \hat{Q}_j\hat{T}_j = RE_j,$$

where $R$ is the residual matrix and $E_j = [0, 0, \cdots, I_b]$ ($I_b$ is the identity matrix of size $b$). Subsequently, the Rayleigh-Ritz procedure is applied by using the extreme eigenpairs $(\hat{\lambda}_i, \hat{\mathbf{u}}_i)$ of $\hat{T}_j$ to obtain the Ritz values $\hat{\lambda}_i$ and Ritz vectors $\hat{Q}_j\hat{\mathbf{u}}_i$ as the approximate eigenpairs $(\bar{\lambda}_i, \bar{\mathbf{u}}_i)$ of $A$. If the residuals $\|A\bar{\mathbf{u}}_i - \bar{\lambda}_i\bar{\mathbf{u}}_i\|$ are small enough, we stop the procedure and output $(\bar{\lambda}_i, \bar{\mathbf{u}}_i)$, $i = 1, \cdots, k$, as the approximate eigenpairs of $A$. The main procedure of block Lanczos is listed in Algorithm 3.

---

**Algorithm 3:** Block Lanczos

**Input**  : $n \times n$ symmetric sparse matrix $A$, rank $k$ and initial matrix $V_0$.
**Output**: The approximate dominant $k$ eigenpairs $(\bar{\lambda}_i, \bar{\mathbf{u}}_i)$, $i = 1, \cdots, k$, of $A$.

1 Initialize block Lanczos: $B_0 = 0$; $Q_0 = 0$; $Q_1 = V_0$
2 **for** $j = 1, 2, \cdots$ **do**
3     $R = AQ_j - Q_{j-1}B_{j-1}^T$   (Let $R$ be orthogonal to $Q_{j-1}$)
4     $A_j = Q_j^T R$   (Obtain $A_j$ by projecting $R$ onto $Q_j$)
5     $R = R - Q_j A_j$   (Let $R$ be orthogonal to $Q_j$)
6     $Q_{j+1}B_j = R$   (QR-factorization of $R$ to obtain $Q_{j+1}$ and $B_j$)
7     Form $\hat{T}_j$ and $\hat{Q}_j$ and compute the top-$k$ eigenpairs $(\hat{\lambda}_i, \hat{\mathbf{u}}_i)$ of $\hat{T}_j$ to obtain the Ritz values $\bar{\lambda}_i = \hat{\lambda}_i$ and Ritz vectors $\bar{\mathbf{u}}_i = \hat{Q}_j\hat{\mathbf{u}}_i$ of $A$.
8     If the residuals $\|A\bar{\mathbf{u}}_i - \bar{\lambda}_i\bar{\mathbf{u}}_i\|$, $i = 1, \cdots, k$, are sufficiently small, then stop and output $(\bar{\lambda}_i, \bar{\mathbf{u}}_i)$ as the approximate eigenpairs.
9 **end**

---

## 6.2 Proof of Theorem 3.1

*Proof.* The proof is based on the $\sin\theta$ theorem in [4]. Let the eigenvectors of the $n \times n$ symmetric matrices $D$ and $D + \Delta$ be $E = [E_0|E_1]$ and $F = [F_0|F_1]$, respectively, where $E_0, F_0 \in \mathbb{R}^{n \times k}$ and $E_1, F_1 \in \mathbb{R}^{n \times (n-k)}$. Then

$$D = E \begin{bmatrix} \Sigma_0^D & 0 \\ 0 & \Sigma_1^D \end{bmatrix} E^T,$$

$$D + \Delta = F \begin{bmatrix} \Sigma_0^{D+\Delta} & 0 \\ 0 & \Sigma_1^{D+\Delta} \end{bmatrix} F^T,$$

$$E_0^T F_0 = U \cos\Theta V^T,$$

where $\Theta$ are the principal angles between $E_0$ and $F_0$. Assume that $\Sigma_0^D \subseteq [a, b]$ and $\Sigma_1^{D+\Delta} \subseteq (-\infty, a - \delta) \cup (b + \delta, \infty)$. Then

$$\|F_1^T E_0\| = \|F_0^T E_1\| = \|\sin\Theta\|.$$

Assume that the eigenvalues $\Sigma_0^D$ lie in some interval, while the eigenvalues $\Sigma_1^{D+\Delta}$ lie in some distance $\eta \geq 0$ from that interval (possibly on both sides of it). Then

$$\|\sin(\Theta(E_0, F_0))\|_2 \leq \frac{\|\Delta E_0\|_2}{\eta} \leq \frac{\|\Delta\|_2}{\eta}, \quad \|\sin(\Theta(E_0, F_0))\|_F \leq \frac{\|\Delta E_0\|_F}{\eta} \leq \sqrt{k}\frac{\|\Delta\|_F}{\eta}.$$

We can set $\eta = \min|\lambda - \hat{\lambda}|$ with $\lambda \in \Sigma_0^D$ and $\hat{\lambda} \in \Sigma_1^{D+\Delta}$. $\qquad\square$

## 6.3  Proof of Theorem 3.3

*Proof.* According to Lemma 6.1,

$$\|A - \bar{U}_k \bar{\Sigma}_k \bar{U}_k^T\|_2 = \|A - QQ^T AQQ^T\|_2 \leq 2\|A - QQ^T A\|_2.$$

where $Q$ is an orthogonal basis for the Krylov subspace $[\Omega, A\Omega, \cdots, A^q A\Omega]$.

First, let $Z = [\Omega, A\Omega, \cdots, A^q A\Omega]$, $Q_B Q_B^T$ is a projector for $B = A^{q+1}$ and $Q_Z Q_Z^T$ is a projector for $Z$. Since we know that $range(A^q A\Omega) \subset range([\Omega, A\Omega, \cdots, A^q A\Omega])$, by Lemma 6.4, we have

$$\|(I - Q_Z Q_Z^T)A\| \leq \|(I - Q_B Q_B^T)A\|.$$

Furthermore, by Lemma 6.2, we have

$$\|(I - Q_B Q_B^T)A\| \leq \|(I - Q_B Q_B^T)B\|^{\frac{1}{(q+1)}}.$$

Next, we need to bound $\|(I - Q_B Q_B^T)B\|^{\frac{1}{(q+1)}}$. According to Lemma 6.3, we have

$$
\begin{aligned}
\|(I &- Q_B Q_B^T)B\|^{\frac{1}{q+1}} \\
&\leq (\|\Sigma_2^B\|^2 + \|\Sigma_2^B \Omega_2^B (\Omega_1^B)^\dagger\|^2)^{\frac{1}{2(q+1)}} \\
&\leq (\|\Sigma_2^B\|^2 (1 + \|\Omega_2^B\|^2 \|(\Omega_1^B)^\dagger\|^2))^{\frac{1}{2(q+1)}} \\
&= \sigma_{k+1}(1 + \|\Omega_2^B\|^2 \|(\Omega_1^B)^\dagger\|^2)^{\frac{1}{2(q+1)}},
\end{aligned}
$$

where $\sigma_{k+1}$ is the $(k+1)$-th largest singular value of $A$.

Next, we show how to bound the error for both $\|\Omega_2^B\|$ and $\|(\Omega_1^B)^\dagger\|$. We already know $\Omega_1 = U_1^T \Omega$, $\Omega_2 = U_2^T \Omega$ and $\Omega \leftarrow \text{diag}(U_{k_1}^{(1)}, U_{k_2}^{(2)}, \ldots, U_{k_c}^{(c)})$, which shows that $\Omega$ is the top-$k$ eigenvectors for the (unperturbed) matrix $D$. With Theorem 3.1, we can now bound $\|\Omega_2^B\|$ and $\|(\Omega_1^B)^\dagger\|$ as

$$\|\Omega_2^B\| = \|\sin\Theta\| \leq \frac{\|\Delta\|}{\eta},$$

$$\|(\Omega_1^B)^\dagger\| = \|\frac{1}{\cos\Theta}\| = \frac{1}{\sqrt{1 - \|\sin\Theta\|^2}} \leq \frac{1}{\sqrt{1 - \frac{\|\Delta\|^2}{\eta^2}}}.$$

As a consequence, we have

$$
\begin{aligned}
\|A - \bar{U}_k \bar{\Sigma}_k \bar{U}_k^T\|_2 &\leq 2\|A - Q_Z Q_Z^T A\| \\
&\leq 2\|(I - Q_B Q_B^T)A\| \\
&\leq 2\|(I - Q_B Q_B^T)B\|^{\frac{1}{(q+1)}} \\
&\leq 2\sigma_{k+1}\left(1 + \frac{\sin^2\theta}{1 - \sin^2\theta}\right)^{\frac{1}{2(q+1)}},
\end{aligned}
$$

where $\theta$ is the largest principal angle of $\Theta$.

$\qquad\square$

Lemmas used in the above proof are listed as follows [7]:

**Lemma 6.1.** *Suppose $A$ is Hermitian and $Q$ is an orthogonal basis, then*

$$\|A - QQ^*AQQ^*\| \leq 2\|A - QQ^*A\| = 2\|(I - QQ^*)A\|.$$

**Lemma 6.2.** *Let $A$ be an $m \times m$ matrix and $\Omega$ be an $m \times \ell$ matrix. Fix a non-negative integer $q$, from $B = A^q A$, and compute the sample matrix $Z = B\Omega$. For an orthogonal basis $Q$ for $Z$,*

$$\|(I - QQ^*)A\| \leq \|(I - QQ^*)B\|^{\frac{1}{q+1}},$$

*where $\|\cdot\|$ represents unitary-invariant norm including the spectral norm and the Frobenius norm.*

**Lemma 6.3.** *Let $A$ be an $m \times n$ matrix with singular value decomposition*

$$A = U\Sigma V^* = U \begin{bmatrix} \Sigma_1 & \\ & \Sigma_2 \end{bmatrix} \begin{bmatrix} V_1^* \\ V_2^* \end{bmatrix}$$

*and fix $k \geq 0$, where the size of $V_1$ and $V_2$ are $n \times k$ and $n \times (n-k)$, respectively. Choose a test matrix $\Omega$ and construct the sample matrix $Y = A\Omega$. Let $\Omega_1 = V_1^*\Omega$ and $\Omega_2 = V_2^*\Omega$. Assuming $\Omega_1$ and $\Omega_2$ has full column rank, the approximation error satisfies*

$$\|(I - P_Y)A\|^2 \leq \|\Sigma_2\|^2 + \|\Sigma_2\Omega_2\Omega_1^\dagger\|^2,$$

*where $\|\cdot\|$ denotes either the spectral norm or the Frobenius norm and $P_Y$ is the projector for $Y$.*

**Lemma 6.4.** *Suppose $range(N) \subset range(M)$. Then, for a matrix $A$, it holds that $\|P_N A\| \leq \|P_M A\|$ and $\|(I - P_M)A\| \leq \|(I - P_N)A\|$, where $P_N$ and $P_M$ are projectors for $range(N)$ and $range(M)$, respectively.*

## 6.4 Performance of MSEIGS with varying cluster quality

To vary the clustering quality, we first cluster the CondMat graph into 4 clusters and then randomly perturb clusters by moving a portion of vertices from their original cluster to another random cluster, which reduces the number of within-cluster edges. Table 4 presents the performance of MSEIGS with different percentages of vertices shuffled. We can see that (1) the quality of clustering influences the performance of MSEIGS, i.e., better quality of clustering implies higher accuracy of MSEIGS; (2) even with poor clustering structure, MSEIGS can still obtain reasonably good approximations.

Table 4: Performance of MSEIGS with varying cluster quality.

| Percent of vertices shuffled | 0% | 20% | 40% | 60% | 80% | 100% |
|---|---|---|---|---|---|---|
| Percent of within-cluster edges | 86.31% | 64.57% | 47.08% | 35.43% | 27.42% | 24.92% |
| Avg. cosine of principal angles | 0.9980 | 0.9757 | 0.9668 | 0.9475 | 0.9375 | 0.9268 |

## 6.5 Experimental settings

All experiments are conducted on computing nodes that have two Intel Xeon E5-2680 (v2) CPUs with either 256 GB or 1 TB of main memory. Our algorithms are implemented in C++ with OpenMP and all methods use Intel Math Kernel Library (MKL) as the underlying BLAS/LAPACK library.

## 6.6 Statistics of datasets used for label propagation

In the experiments, we use the RBF kernel $W_{ij} = \exp(-\gamma\|\mathbf{x}_i - \mathbf{x}_j\|^2)$ to measure the similarity between samples $i$ and $j$. In Table 5, we present statistics and parameters of the dataset used for label propagation in Section 4.2. The parameters are chosen by cross-validation.

Table 5: Statistics of datasets used for label propagation.

| Dataset | # of training points | # of test points | # of classes/labels | dimension | $\gamma$ | $\alpha$ |
|---|---|---|---|---|---|---|
| Delicious | 12,920 | 3,185 | 983 | 500 | $10^{-1}$ | 0.99 |
| Aloi | 10,000 | 98,000 | 1,000 | 128 | $10^{-7}$ | 0.99 |

## 6.7 Statistics of datasets used for inductive matrix completion

In Table 6, we give a summary of the datasets used for inductive matrix completion in Section 4.3. Note that $r$ is the rank of $W$ and $H$ in IMC and we set the regularization parameter $\lambda = 0.1$, which are chosen by cross-validation. For the Amazon and LiveJournal datasets, we randomly sampled items (affiliations) with at least 10 users.

Table 6: Statistics of datasets used for inductive matrix completion.

| Dataset | # of users | # of items | # of ratings in $R$ | # of links in $A$ | $r$ |
|---|---|---|---|---|---|
| Flixster | 1.0M | 48.8K | 8.2M | 11.8M | 100 |
| Amazon | 334.8K | 73.2K | 2.7M | 1.9M | 200 |
| LiveJournal | 4.0M | 2.0K | 2.4M | 69.4M | 100 |