[Reviews · NeurIPS 2014]

Submitted by Assigned_Reviewer_6

The paper addresses the problem of estimation of eigenvalues and eigenvectors of large sparse graph matrices. It uses a nice divide-and-conquer approach to obtain better estimates of top-k eigenspace. Such an estimate can be used in several classic tasks, such as link prediction and recommender systems.

The paper is build upon a divide-and-conquer method, taking advantage of the particular characteristics of some graphs (e.g. social networks): sparsity and cluster structure.

The theoretical part is strong, with theoretical bounds provided. The divide part uses a decomposition of the adjacency matrix as a sum of a block-diagonal matrix and an almost zero non-diagonal matrix. A spectral decomposition of block matrices is done, and the k first eigen values are chosen among the resulting union of eigenvalues.

In the conquer step, the corresponding eigenvectors are used as initialization step in a block-Lanczos (or random SVD) method on the whole matrix.

Theoretical bounds are provided, with respect to the norm of the remaining matrix (the non-block one).

Experiments show that the method is quite accurate for the eigenspace identification, compared to classic spectral methods (randomized SVD or Lanczos).

In tasks such as label propagation and matrix completion, the methods performs as well as other classical methods.

The paper is well written. It uses quite sophisticated methods and the theoretical part is solid.

The main weakness of the paper is that the tasks that are presented as highly dependent on the accuracy of eigenspace estimation (link prediction, etc.) reveals to be not so dependent: the experiments show that the method presented in the paper is much more accurate than previous state-of-the-art in eigenspace estimation, but this does not implies significantly better results in the label propagation or matrix completion tasks.

In other words, The goal of the method (a good estimate of the eigenspace) is reached, but it appears that it has no effect on the tasks that it was supposed to solve (except for the computing time).

Also, the method (and the bound) should highly depend on the quality of the first step (the graph clustering step), and it would have been nice to have an idea of the accuracy of the method with respect to this first step.

However, the method is original and could be used in other spectral decomposition problems (provided that they are sparse and likely to be clustered).

Pros:
- nice divide-and-conquer algorithm
- solid theoretical part
- theoretical bounds
- experiments show that the method reaches good accuracy

Cons:
- for real-world tasks, the algorithm does not perform significantly better that previous methods
- dependency between first clustering step and quality of the solution is not studied.
Summary: The method described in the paper seems original, and could be promising, although the experiments show only marginal improve wrt other classical methods.

Submitted by Assigned_Reviewer_12

The authors propose a new approach to compute the $k$ dominant eigenvector/value pairs of large sparse graphs (i.e. positive and symmetric adjacency matrices). Their methods consist of (i) recursively clustering the graph (divide), and (ii) at each step (while going back from leafs to root) using the dominant eigenvectors of the clusters as initialisations to a block Laczos algorithm which approximate the dominant eigenvector/value pairs of the cluster from the next level (conquer). Theoretical guarantees are provided and the method is experimentally validated on two real machine learning applications (label propagation and inductive matrix completion) where it outperforms state of the art methods.

Quality:
The theoretical guarantees and experiments are convincing. Two points that may need clarification:
- You didn't comment on the choice of the number $r$ of top eigenvector/value pairs to extract from each cluster. However it seems that its choice could be tedious: if $r$ is too small we may miss a dominant eigenvector of the original graph in one of the clusters, and choosing $r$ too big may lead to bad time complexity. (if $r$ is indeed a parameter of your algorithm you should put it in input of Algorithm 1)
- You provided theoretical guarantees for the one-level spectral decomposition but not for the multi-level one. Can you comment on how this multi-level strategy would affect your theoretical results?

Clarity:
The paper is well written and easy to follow.

Originality:
The idea seems original.

Significance:
Fast and efficient methods to compute the spectral decomposition of large matrices are of great interest to the machine learning community. The fact that the proposed method can easily be parallelised and that an early termination strategy is presented is nice.

Summary: The paper is well written, the idea seems original and both the theoretical analysis and the experiments are convincing.

Submitted by Assigned_Reviewer_22

The paper proposes a technique for computing the eigenvectors of large matrices
arising from graphs, by exploiting the clustering structure that usually appears
in multiple levels of such objects. Graph clustering is first applied followed by
spectral decomposition on each cluster. This serves as an initial approximation
for a block Lanczos algorithm on the complete graph.

The paper is well written and computing eigenvectors of huge matrices has many applications in machine learning. The contribution of this paper quite straightforward and the experiments show the diversity of applications that this work benefits.

I think the paper could do a better job putting the existing
work in context. In particular I am very curious as to why randomized SVD is such a poor contender in this paper. It might be explained by performing unnecessarily too many passes over the dataset (usually more than two passes do not add anything more). Minimizing the number of passes is the most relevant quantity in big data applications and in this light of this, it is unclear whether the presented results would hold once we move to a setting where the data does not fit in memory and has to be streamed from
disk (or even when the data lives in a distributed file system). Randomized SVD has
gained popularity because its data access pattern enables such scenarios.

At the same time RSVD does not need any preprocessing while the proposed method needs to run a graph clustering algorithm and in the experiments it is not even clear what is the proportion of time spent clustering versus executing Lanczos. It is also unclear whether the graph clustering algorithms would work efficiently as we move to datasets that don't fit to a single machine's memory.

Other notes:
-Theorems and Lemmas seem to depend on results from Stewart and Sun though no reference is given
-Line 418: MSIEGS -> MSEIGS
Summary: The paper proposes a smart initialization for block Lanczos of matrices arising from natural graphs. It would meet the bar if it was clear
a) why it beats RSVD so much
b) what happens when datasets don't fit in memory
c) what is the running overhead from clustering.
--Update after author response:
The authors provided good explanations for (a), plausible ways to handle (b) and concrete timings for (c). Therefore, I have substantially increased my score.
Author Feedback
Author rebuttal: We thank reviewers for their comments/suggestions.

To Reviewer_12:
1:The choice of r eigenpairs extracted from each cluster.

As in lines 191 to 193 in the paper, r is proportional to the Frobenius norm of each cluster. The intuition is that larger clusters tend to have more influence over the spectrum of the entire matrix. Suppose we target top-100 eigenpairs of A, we partition A into 3 clusters where the Frobenius norm ratio of each cluster is 0.2, 0.3, 0.5. First we decide the number of eigenvectors to get from the child clusters using a small amount of oversampling(0.2*k), and then distribute the 120 eigenvectors to each cluster. So r for each cluster is 24, 36 and 60.

2:How multi-level strategy affects the theoretical results.

The multi-level strategy is used to speed up the computation when some clusters are still too large to compute their eigendecompositions efficiently, so we further divide the clusters to smaller ones. Thus multi-level MSEIGS is an approximation of single-level MSEIGS. The theoretical guarantees may be generated to multi-level MSEIGS by accumulating errors from each level of the hierarchy, which potentially implies a cumulative error bound for the multi-level case as compared to the single-level case.

To Reviewer_22:
3:why MSEIGS beats RSVD so much?

With the same number of iterations, block Lanczos and RSVD have the same number of passes over the graph. The number of passes (or iterations) required is related to the desired accuracy and decay of the eigenvalues. If the decay is slow or highly accurate approximation is needed, we usually need several iterations(>5 iterations).

As described from lines 176 to 189 in the paper, block Lanczos uses the j-th Krylov subspace, span(V,A*V,...,A^j*V), at the j-th iteration, while RSVD restricts to a smaller subspace span(A^j*V) discarding information from previous iterations. So block Lanczos outperforms RSVD with the same number of iterations. For example, on the CondMat dataset with target rank 100, block Lanczos needs 7 iterations, while RSVD takes more than 10 iterations to achieve similar accuracy(0.99). Furthermore, MSEIGS needs only 5 iterations by using a better initialization than block Lanczos. Figures 1(b) and 1(c) show that MSEIGS is more accurate than both block Lanczos and RSVD with a fixed number of iterations.

4: What happens when data don't fit in memory or in a distributed file system?

In this paper, we mainly focus on approximating eigendecomposition under the multi-core shared-memory setting. Dealing with graphs that cannot fit into memory is one of our future research directions. We believe that with careful implementation of MSEIGS, it can also be efficient in streaming and distributed settings. Here we briefly outline one way to implement MSEIGS for such cases.

Let us start with the single-level MSEIGS. First we can apply streaming graph clustering algorithms[1] to generate c clusters such that each cluster can fit into memory. The graph can be organized into a c-by-c block matrix where A_ij consists of links between cluster i and j. Then we load the subgraphs A_11,...,A_c1 and compute eigenvectors U_1 of A_11. After that, we multiply U_1 with the previously loaded subgraphs. Then we repeatedly apply this procedure for each cluster and obtain AU where U=diag(U_1,...,U_c). For the multi-level case, we can apply MSEIGS under the shared-memory setting when we load each A_ii into memory.

For the distributed case, we can use existing distributed graph clustering algorithms(ParMetis) and then compute each cluster's eigenpairs independently in each machine. For the top level, we apply distributed Lanczos algorithms[2].
While this is future work, the above ideas show the potential of the ideas presented in the paper.

5: What is the running overhead from clustering. How graph clustering algorithms works when data cannot fit into memory.

Compared with block Lanczos step, the overhead of clustering is very low, usually less than 10% of the total time. For the FriendsterSub dataset with 10M nodes and 83M edges, to achieve 0.99 accuracy, MSEIGS takes 1825 secs where clustering takes only 80 secs. MSEIGS is a framework, so we can apply any graph clustering algorithm including distributed/streaming algorithms, such as ParMetis, Fennel[1] and GEM([25] in the paper), to generate the partitions when the data cannot fit into memory. We will add these results/discussions to the paper.

[1] C. Tsourakakis, C. Gkantsidis, B. Radunovic and M. Vojnovic. Fennel: Streaming graph partitioning for massive scale graphs. WSDM, 2014.
[2] M. R. Guarracino, F. Perla and P. Zanetti. A parallel block Lanczos algorithm and its implementation for the evaluation of some eigenvalues of large sparse symmetric matrices on multicomputers. AMCS 16(2):241-249, 2006.

To Reviewer_6:
6:The comparison of MSEIGS with other methods for real-word tasks.

MSEIGS is faster and more accurate than other methods for approximating eigenpairs. To test the performance of MSEIGS for various machine learning tasks, we run each algorithm to achieve similar accuracy and compare the computational time. So all methods have similar accuracy, with different running times. To achieve similar accuracy, MSEIGS is much faster. For instance, on the Aloi dataset, MSEIGS is about 5 times faster and MSEIGS-Early is almost 20 times faster than Matlab's EIGS.

7: Dependency between clustering step and quality of solution.

Better quality of clustering(more within-cluster links) implies higher accuracy of MSEIGS. To vary the clustering quality, we cluster the CondMat graph into 4 clusters and then randomly perturb clusters by moving a portion of vertices from their original cluster to another random cluster to reduce within-cluster links. The result is:
Ratio of vertices shuffled: 0 0.2 0.4 0.8 1
Ratio of within-cluster links: 0.8631 0.6457 0.4708 0.2742 0.2492
Accuracy of MSEIGS: 0.9980 0.9757 0.9668 0.9375 0.9268